# GDNF, A Neuron-Derived Factor Upregulated in Glial Cells during Disease

**DOI:** 10.3390/jcm9020456

**Published:** 2020-02-07

**Authors:** Marcelo Duarte Azevedo, Sibilla Sander, Liliane Tenenbaum

**Affiliations:** Laboratory of Molecular Neurotherapies and NeuroModulation, Center for Neuroscience Research, Lausanne University Hospital, CHUV-Pavillon 3, av de Beaumont, CH-1010 Lausanne, Switzerland; Marcelo.Duarte-Azevedo@chuv.ch (M.D.A.); sibilla.sander@chuv.ch (S.S.)

**Keywords:** glial-cell-line-derived neurotrophic factor, microglia, astrocyte, neuroinflammation, rearranged during transfection, GDNF family receptor alpha 1, gene therapy, Parkinson’s disease

## Abstract

In a healthy adult brain, glial cell line-derived neurotrophic factor (GDNF) is exclusively expressed by neurons, and, in some instances, it has also been shown to derive from a single neuronal subpopulation. Secreted GDNF acts in a paracrine fashion by forming a complex with the GDNF family receptor α1 (GFRα1), which is mainly expressed by neurons and can act in *cis* as a membrane-bound factor or in *trans* as a soluble factor. The GDNF/GFRα1 complex signals through interactions with the “rearranged during transfection” (RET) receptor or via the neural cell adhesion molecule (NCAM) with a lower affinity. GDNF can also signal independently from GFRα1 by interacting with syndecan-3. RET, which is expressed by neurons involved in several pathways (nigro–striatal dopaminergic neurons, motor neurons, enteric neurons, sensory neurons, etc.), could be the main determinant of the specificity of GDNF’s pro-survival effect. In an injured brain, de novo expression of GDNF occurs in glial cells. Neuroinflammation has been reported to induce GDNF expression in activated astrocytes and microglia, infiltrating macrophages, nestin-positive reactive astrocytes, and neuron/glia (NG2) positive microglia-like cells. This disease-related GDNF overexpression can be either beneficial or detrimental depending on the localization in the brain and the level and duration of glial cell activation. Some reports also describe the upregulation of RET and GFRα1 in glial cells, suggesting that GDNF could modulate neuroinflammation.

## 1. Introduction

Glial cell line-derived neurotrophic factor (GDNF) has been isolated from the conditioned media of a rat glioma cell line on the basis of its trophic activity towards primary cultures of dopaminergic neurons [1].

Following the administration of GDNF family ligands (GFL) in animals, neurorestorative effects have been demonstrated in models of several neurological diseases [2,3,4]. However, in cases of high doses and long-term administration, aberrant sprouting and negative feedback effects on neurotransmitter homeostasis have been observed [5,6,7,8].

Two members of the GFL, GDNF and neurturin, have been evaluated in clinical trials for the treatment of Parkinson’s disease (PD) [9,10]. Although positron-emission tomography (PET) scan imaging has evidenced clear functional improvements [11,12], and post-mortem analysis has demonstrated neuronal sprouting at the site of delivery [13], clinical outcomes have been disappointing. Therefore, better knowledge of GDNF’s mechanism of action in vivo in complex neuronal circuits is urgently needed in order to revisit the relevant clinical paradigms.

GDNF is mainly expressed during development and is involved in neuronal specification [14,15,16,17]. In a healthy adult brain, GDNF expression decreases and is restricted to specific regions: the cortex, hippocampus, striatum, substantia nigra (GDNF mRNA expression in substantia nigra dopaminergic neurons has been observed in [18] but not in [19]. This discrepancy could be explained by the differences between the probes used for in situ hybridization, the sensitivity of the method (digoxygenin-labeled versus radiolabeled probes), and the sex of the animals (female versus male)), thalamus, cerebellum, and spinal cord [20,21,22]. GDNF is a secreted factor [23] that is primarily expressed by neurons and acts mainly on neurons expressing the “rearranged during transfection” (RET) receptor. In a few instances, it has also been demonstrated to derive from a specific neuronal subpopulation [18,20]. For example, in the striatum, only interneurons [24] (mainly parvalbumin-expressing interneurons) [18,19] express GDNF. Interestingly, the role of parvalbumin neuron-derived GDNF in the maintenance of dopaminergic neurons in the adult brain has been demonstrated using conditional knock-out mice [25]. Thus, in a healthy rodent brain, GDNF appears to be a neuron-derived neurotrophic factor rather than a glia-derived neurotrophic factor. The pattern of GDNF expression is different in a diseased brain. Indeed, de novo GDNF expression in glial cells has been described in numerous models of diseases, usually concomitantly with neuroinflammation.

In this review, we will discuss the dual role of GDNF upregulation in glial cells during neurodegeneration and repair. Other reviews covering the expression of neurotrophic factors by glial cells already exist [26,27]. Contradictory conclusions often arise from data obtained in vivo and in vitro (see discussion in reference [28]). In the present review, for clarity, we will focus on the expression of GDNF and its receptors in the nervous system in vivo, except for some aspects that have only been addressed in vitro.

## 2. Glial Cells Express GDNF during Disease

GDNF is expressed by neurons in developing [29,30] and adult nervous systems [18,20,22,29,30]. The expression patterns of GDNF have been confirmed using genetically modified mice [31] and do not match the astrocyte distribution, as revealed using anti-glial fibrillary acid protein (GFAP) antibodies [20]. Therefore, the name given to this neurotrophic factor can be misleading. Contrasting most publications that have failed to demonstrate GDNF glial expression in a healthy brain, Ubhi et al. have shown that a small proportion (around 6%) of GDNF-expressing cells are glial [32].

Numerous studies have evidenced the de novo expression of GDNF by glial cells in an injured brain (see Table 1). In disease models, neuroinflammation can upregulate GDNF expression in activated astrocytes [33,34,35,36,37,38,39,40], microglia and infiltrating macrophages [33,36,41,42,43,44,45], nestin/GFAP-positive reactive astrocytes [46], and ionized calcium-binding adapter molecule 1 (Iba1)-positive neuron-glial antigen 2 (NG2) expressing macrophage/microglia-like cells [47]. This disease-related GDNF overexpression can be either beneficial [47,48] or detrimental depending on the age of the animal [49], the length of the neuroinflammatory response [36,50], and the type of glial cells activated [28].

GDNF produced by activated microglia/macrophages can lead to a repair of central nervous system (CNS) injuries. After striatal mechanical injury [41,42] and spinal cord injury [43], activated microglia and macrophages express GDNF, thereby inducing axonal sprouting and locomotor improvements. Indeed, in the latter model [51], inhibition of GDNF expression using antisense oligonucleotides drastically reduced axonal sprouting [51]. In the 6-hydroxydopamine (6-OHDA) PD model, around 60% of surviving tyrosine hydroxylase (TH)-positive neurons were located near NG2 cells that expressed GDNF [47]. In the 1-methyl-4-phenyl-1,2,3,6-tetrahydropyridine (MPTP) PD model, cinnamon-induced neuroprotection was shown to be mediated by astrocytic GDNF overexpression in the substantia nigra. The role of astrocyte-derived GDNF in this model was demonstrated by the absence of neuroprotection in knock-out mice lacking GDNF expression in astrocytes [39].

In addition, macrophage-mediated GDNF delivery based on transduced hematopoietic stem cell (HSC) transplantation has successfully rescued nigral dopaminergic neurons and improved motor function in a PD mouse model [52,53]. Chen et al. showed that these gene-modified macrophages/microglia expressing GDNF are recruited to the areas affected by dopaminergic neuron loss and are present in the immediate surroundings of tyrosine hydroxylase (TH) positive cells.

However, glial GDNF overexpression could be a double-edged sword. Indeed, after a mechanical injury of the striatum, GDNF-induced axonal sprouting failed to cross over the wound edge [41]. This dual effect of local GDNF overexpression was also observed in a spinal cord repair paradigm in which a transplanted nerve root genetically modified with a lentiviral vector expressing GDNF in Schwann cells stimulated the regeneration of motor neuron axons locally but not beyond the lesion [6]. It should be noted, however, that the deleterious effect of local GDNF overexpression was not limited to the glia, since long-lasting GDNF overexpression by neurons can also lead to aberrant sprouting in the case of ectopic localization [5,54].

Transgenic mice expressing GDNF, either from an endogenous locus or from a GFAP promoter, revealed that astrocytic-derived GDNF overexpression is responsible for TH downregulation, decreased dopaminergic neurotransmission, and motor deficits [28].

## 3. Glial Cells Express GDNF Receptors during Disease

GDNF forms a complex with its primary receptor, GFRα1, which can be membrane-bound or released in a soluble form [55,56]. The existence of a soluble form of GFR 1 has been shown in the primary cultures of neurons. The GDNF–GFRα1 complex then binds the RET [22,57] present on neuronal cell bodies and terminals of several different pathways, such as nigro–striatal dopaminergic neurons [58,59], spinal motor neurons [60], noradrenergic neurons of the locus coeruleus [61], enteric neurons [62], and sensory neurons [63]. The GDNF–GFRα1 complex can also bind to and induce signaling through the neural adhesion molecule, NCAM [64,65,66,67]. GDNF can furthermore directly interact with the heparin sulfate proteoglycan, syndecan-3 [68,69,70]. Interestingly, GDNF binding to heparan sulfate has been shown to be beneficial for the protection of dopaminergic neurons in the 6-OHDA rat model of PD [71].

The upregulation of GFRα1 and RET has been reported in glial cells under pathological conditions [34,35,72,73,74] (see Table 2).

RET was shown to be expressed in microglia in the brain tissue of patients with PD but not in healthy controls [72]. RET and its phosphorylated form (pRET) were also gradually increased in the microglia during disease progression in a transgenic mouse model of amyotrophic lateral sclerosis (ALS) [74,75]. In parallel, RET expression was decreased in motor neurons. These data suggest that motor neurons die due to a lack of response to neurotrophic factors or due to an excess of neurotoxic compounds derived from activated microglia.

On the other hand, increased neuronal survival concomitant with RET activation in the microglia was described in hippocampal slices treated with the excitotoxin, N-methyl-D-aspartate (NMDA) [73].

Excitatory amino acids induced GDNF and GFRα1 but not RET de novo expression via astrocytes in the striatum following treatment with quinolinate or kainate [34,35]. Similarly, after a mechanical lesion of the spinal cord in adult rats, GDNF and GFRα1 were upregulated in the astrocytes [76].

## 4. Conclusions and Further Prospects

The physiological role of GDNF has been subject to debate. In fact, embryonic *Gdnf* knock-out mice result in neonatal death due to renal agenesis [79,80]. To better understand the importance of GDNF for the survival of catecholaminergic neurons in vivo, in the adult brain, conditional knock-out of *Gdnf* has been carried out, resulting in controversial conclusions. In one study, GDNF has been highlighted as an essential factor for the survival of these neurons [81]. Furthermore, a recent article has shown that GDNF is necessary for the maintenance of mesencephalic catecholaminergic neurons, also on the basis of a conditional knock-out of *Gdnf* gene specifically in parvalbumin-positive neurons of adult mice [25]. In contrast, another study has suggested that GDNF expression was dispensable for the survival of catecholaminergic neurons [82]. It should be noted that none of the conditional *Gdnf* knock-out studies have achieved complete *Gdnf* gene ablation. Interestingly, GDNF overexpression from the native locus, i.e., in parvalbumin-positive neurons, leads to an increased number of dopaminergic neurons in the substantia nigra, increased dopamine transporter (DAT) activity, increased dopamine neurotransmission, and improved motor behavior [19].

From a mainly neuron-derived secretion with specific neurotrophic action during development, GDNF becomes a glia-derived factor that can rescue neurons but also possibly support glial cell activation during neuroinflammation [35,75] (See Figure 1).

In pathological conditions, astrocytic GDNF expression has been reported and shown to be beneficial [39,83]. However, transgenic mice overexpressing GDNF in astrocytes present adverse effects such as TH downregulation, decreased dopamine neurotransmission, and motor deficits [28]. Taken together, these results suggest that prolonged astrocytic overexpression is harmful.

On other hand, microglial GDNF expression has been reported to have beneficial effects in Parkinson’s disease and other animal models of inflammation [36,41,42,43,47,51]. Moreover, after transplantation, genetically-modified hematopoietic stem cells expressing GDNF migrate to the areas affected by dopaminergic neuron loss in the close surroundings of remaining TH positive cells and achieve neuroprotection and motor improvements [52,53]. To our knowledge, adverse effects of microglial GDNF expression have not been reported.

Activated microglia and astrocytes exist in different states, which can be neuroprotective [41,42,43,84,85] or neurotoxic [50,86]. Numerous studies suggest that acute neuroinflammation resulting in the phagocytosis of dead cell debris is beneficial. In contrast, continuous neuroinflammation becomes deleterious due to the high levels of cytokines, reactive oxygen species, and nitrogen species, which are toxic to neurons [87]. The attenuation of a sustained neuroinflammatory response actually increases neuronal survival [75].

Neuroinflammation has been shown to induce the de novo expression of GDNF in glial cells [36,43,88], possibly via the nuclear factor-kappa B (NF-κB)-responsive elements present in the GDNF promoter [89,90,91]. In turn, GDNF, after binding to GFRα1 (also upregulated in disease), could possibly increase the survival of the activated microglia through the activation of RET signaling. It is, therefore, tempting to hypothesize that in situations where the neuroinflammatory process becomes uncontrollable, disease-induced GDNF could contribute to perpetuate microglial activation. Astrocytes were shown to express GDNF and GFRα1, but not RET, under pathological conditions [34]. However, since GFRα1 is a diffusible factor, GDNF could induce trophic signaling in other cell types expressing RET or NCAM.

GDNF and neurturin have been proposed to be therapeutic agents for PD [9,10,11]. Although functional improvements were observed by PET scan imaging [11,12,92], and fiber sprouting was observed in post-mortem samples [13,93], the clinical benefits were very modest.

The emerging picture of the deleterious effects of long-term uncontrolled GDNF overexpression suggests that clinical benefits could have been reduced by aberrant neurotrophic activity inhibiting bona fide neuronal circuit repair.

In gene therapy paradigms using GFL [9,94], our assumption is that transgene expression should be controlled to avoid aberrant sprouting and the perpetuation of neuroinflammatory processes, which can become deleterious. Clinically-acceptable genetic switches are becoming available and could improve the outcomes of future clinical trials using GFL [8,95,96,97,98,99,100,101].

## Figures and Tables

**Figure 1 jcm-09-00456-f001:**
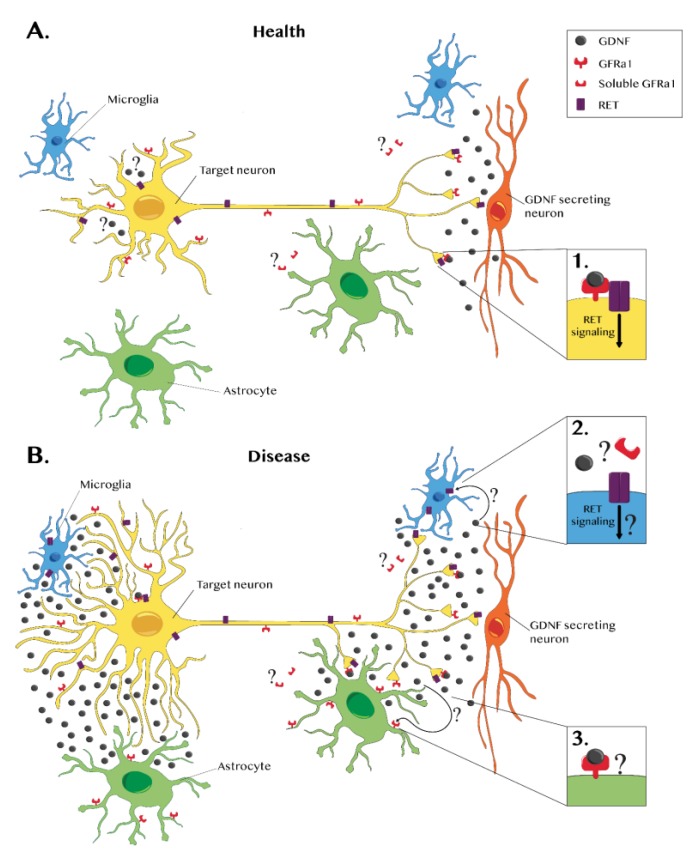
Glial cell line-derived neurotrophic factor (GDNF) and GDNF receptors in a healthy brain and during disease. (**A**) In a healthy nervous system, GDNF expression is mainly neuronal (red). GDNF forms a complex with the GDNF family receptor alpha 1 (GFRα1), which is present in the neuronal membrane. This complex binds to RET, a transmembrane receptor, triggering an intracellular signaling cascade that promotes survival (see inset 1). A few in vitro studies have reported that GFRα1 also exists in a soluble form, suggesting that GDNF can have broader effects. However, these data lack in vivo confirmation. Some neurons express GDNF and its receptors. Therefore, a possible autocrine GDNF effect should not be excluded even though it has not been demonstrated yet. (**B**) Several studies report that during disease, glial cells can also express GDNF. Glial GDNF expression can promote survival and axonal growth, but sustained GDNF overexpression or ectopic GDNF expression can lead to aberrant sprouting. In pathological cases, the microglia (blue) express RET but not GFRα1, suggesting that RET signaling may occur in a GDNF-independent manner or through a GFRα1 soluble form (see inset 2). In disease conditions, GFRα1 is upregulated in astrocytes (green), but there is no evidence of RET expression (see inset 3). Further investigation is required to establish the effects of GDNF–GFRα1 astrocytic interactions.

**Table 1 jcm-09-00456-t001:** Non-neuronal glial cell line-derived neurotrophic factor (GDNF) expression during disease.

Cell Type	Disease Model	Methods Used	References
**Macrophages/Microglia**	Striatal mechanical injury	ISH + immunohistochemistry	[41,42,44]
	Experimental autoimmune neuritis	Double immunofluorescence	[45]
	LPS-induced inflammation	Double immunofluorescence	[36,43]
**Astrocytes**	Quinolic acid lesion	Double immunofluorescence	[34,35]
	LPS-induced inflammation	Double immunofluorescence	[36]
	6-OHDA	ISH + immunofluorescence	[37]
		Double immunofluorescence	[38]
	MPTP	Double immunofluorescence	[39]
	Spinal cord ischemia	Double immunofluorescence	[40]
**Nestin-Expressing Reactive Astrocytes**	MPTP	Double immunofluorescence	[46]
**Microglia-Like NG2-Expressing Cells**	6-OHDA	Double immunofluorescence	[47]

ISH, in situ hybridization; 6-OHDA, 6-hydroxydopamine; LPS, lipopolysaccharide: MPTP; 1-methyl-4-phenyl-1,2,3,6-tetrahydropyridine; neuron-glial antigen 2 (NG2).

**Table 2 jcm-09-00456-t002:** Upregulation of the glial cell line-derived neurotrophic factor (GDNF) receptors in activated glial cells.

Receptor	Cell Type	Disease/Lesion	Methods	Reference
GFR**α**1	Astrocytes	Rat striatum treated with quinolinic acid	Double immunofluorescence	[35]
		Rat striatum treated with quinolinic acid or kainic acid	GFRα1 immunoreactivity, morphology	[34]
		Spinal cord mechanical injury	GFRα1 immunoreactivity, localization in white matter, morphology.	[76]
RET	Microglia	Human PD and aging	Single immunohistochemistry, morphology	[72]
RET, pRET	Microglia	ALS transgenic mice	Double immunofluorescence.	[74]
pRET	Microglia?	Rat hippocampal slices treated with NMDA and exogenous GDNF	Immunofluorescence combined with isolectin IB4 *	[73]

GFRα1, GDNF family receptor alpha 1; PD, Parkinson’s disease; ALS, amyotrophic lateral sclerosis; NMDA, N-methyl-D-aspartate; rearragend during transfection (RET); pRET, phosphorylated RET. * IB4, an isolectin widely used to label microglial cells in vitro [77], was shown to directly interact with RET—an observation that calls into question the identification of IB4-labeled cells in vivo [78].

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
