# Peer review of "GDNF, A Neuron-Derived Factor Upregulated in Glial Cells during Disease"

_jcm, 2020, doi:10.3390/jcm9020456_

Round 1

Reviewer 1 Report

This is a review report on roles of GDNF and the receptor in the central nervous system. Authors introduced that various pathological conditions induce GDNF expression in glial cells including activated astrocytes and microglia, macrophages, neural stem cells and NG2-expressing progenitor cells, although the expression is shown in neurons under physiological condition.

This review is interesting, however three points shown below should be reconsidered:

Ref. 44 indicates nestin-expressing reactive astrocytes. Are those cells neural stem cells? Such description should be reconsidered. Ref. 45 indicates microglia-like NG2 glial cells expressing NG2 and Iba1. Are those cells glial progenitor cells? Such a description should be reconsidered. Figure 1 is not so understandable for readers. Roles of GDNF released from glial cells have to be described in more detail and in a comprehensible fashion.

Author Response

This is a review report on roles of GDNF and the receptor in the central nervous system. Authors introduced that various pathological conditions induce GDNF expression in glial cells including activated astrocytes and microglia, macrophages, neural stem cells and NG2-expressing progenitor cells, although the expression is shown in neurons under physiological condition.

This review is interesting, however three points shown below should be reconsidered:

Ref. 44 indicates nestin-expressing reactive astrocytes. Are those cells neural stem cells? Such description should be reconsidered.

We thank the reviewer for his valuable comment. Indeed, ref.44, now ref.46 describes nestin-expressing reactive astrocytes and not neural stem cells. We have modified the manuscript accordingly (see lines 71/72).

Ref. 45 indicates microglia-like NG2 glial cells expressing NG2 and Iba1. Are those cells glial progenitor cells? Such a description should be reconsidered.

We thank the reviewer for his valuable comment. Indeed, ref.45, now ref 47 describes macrophage/microglia-like NG2-expressing cells. We have modified the manuscript accordingly (see lines 72/73).

Figure 1 is not so understandable for readers.

We have improved Figure 1 to make it more easily understandable:

-the interactions between GDNF and its receptors are now shown within insets which contain enlargements.

-some symbols (e.g. the circles representing GDNF) have been enlarged.

-the legend better describes the demonstrated and hypothetical interactions between GDNF and its receptors on neuronal and glial cells.

Roles of GDNF released from glial cells have to be described in more detail and in a comprehensible fashion.

The roles of GDNF released from glial cells have been described in more details in the main text (lines 76-85) and in the discussion. In particular, tools that have been used to demonstrate GDNF’s role (knock-out mice, overexpressing mice, antisense oligonucleotides) are described.

Reviewer 2 Report

This review of GDNF upregulation in glial cells during disease is reasonably comprehensive but does not demonstrate critical insights or critical analysis of the literature. 

The manuscript would benefit greatly from the editorial skills of a native English speaker. 

Author Response

This review of GDNF upregulation in glial cells during disease is reasonably comprehensive but does not demonstrate critical insights or critical analysis of the literature

We have provided a more critical analysis of the literature in the discussion (lines 136 to 162 and following).

We have discussed the data showing that GDNF is required for the maintenance of adult dopaminergic neurons and opposed them to other data showing that long-term unregulated neuronal GDNF overexpression via viral vectors can provoke adverse effects. We have have also opposed controversial data showing beneficial effects of astrocytic GDNF expression and those showing deleterious effects of long-term GDNF overexpression. Finally, we have gathered data from several different paradigms and systems indicating that microglia secreting GDNF are found in the immediate vicinity of the lesioned neurons and support their survival. 

The manuscript would benefit greatly from the editorial skills of a native English speaker. 

The manuscript has been edited by MDPI English editing service. Modifications are enlighted in the revised manuscript.

Reviewer 3 Report

In this manuscript the Authors reviewed early and more recent literature studies regarding the role of GDNF in healthy adult brain and injured brain. The issue addressed in this manuscript is of interest in the field of neuroscience. The Authors provided a critical discussion on this theme and literature data are described in a satisfactory way.

This review is very interesting, well written and will be of interest to researchers and clinicians working on disease-related GDNF. From the reviewer’s point of view, minor revision is needed to improve the article and make it acceptable for publication. In particular the Authors should introduce more recent literature of the last year.  

Author Response

In this manuscript the Authors reviewed early and more recent literature studies regarding the role of GDNF in healthy adult brain and injured brain. The issue addressed in this manuscript is of interest in the field of neuroscience. The Authors provided a critical discussion on this theme and literature data are described in a satisfactory way.

This review is very interesting, well written and will be of interest to researchers and clinicians working on disease-related GDNF.

From the reviewer’s point of view, minor revision is needed to improve the article and make it acceptable for publication.

Numerous minor revisions have been introduced in the manuscript according to other reviewer’s comments. We hope that these are satisfactory.

In particular the Authors should introduce more recent literature of the last year.  

Indeed, we had missed important articles i) describing a very interesting strategy delivering GDNF in parkinson’s disease models via the autologous transplantation of genetically-modified hematopoietic stem cells which migrate to the substantia nigra and express microglial markers (refs.52 and 53); ii) showing that GDNF is important for the maintenance of adult dopaminergic neurons (ref.25).

We apologize if we are still missing important recent work..